# Clinical Implication of Concurrent Amplification of MET and FGFR2 in Metastatic Gastric Cancer

**DOI:** 10.3390/biomedicines11123172

**Published:** 2023-11-28

**Authors:** Seonggyu Byeon, Jaeyun Jung, Seung Tae Kim, Kyoung-Mee Kim, Jeeyun Lee

**Affiliations:** 1Division of Hematology-Oncology, Department of Internal Medicine, Ewha Womans University Seoul Hospital, Seoul 07804, Republic of Korea; bichon3@naver.com; 2Division of Hematology-Oncology, Department of Medicine, Samsung Medical Center, Sungkyunkwan University School of Medicine, Seoul 06351, Republic of Korea; 3Department of Pathology and Translational Genomics, Samsung Medical Center, Sungkyunkwan University School of Medicine, Seoul 06351, Republic of Korea; kkmkys@skku.edu

**Keywords:** MET, FGFR2, co-amplification, gastric cancer

## Abstract

Background: c-mesenchymal epithelial transition factor receptor (c-MET) and fibroblast growth factor receptor 2 (FGFR2) amplification have been identified as factors associated with advanced stage and poor prognosis in gastric cancer (GC). While they are typically considered mutually exclusive, concurrent amplifications have been reported in a small subset of GC patients. Methods: in this retrospective study, we analyzed the clinical outcomes of GC patients with MET and FGFR2 amplification using the next-generation sequencing (NGS) database cohort at Samsung Medical Center, which included a total of 2119 patients between October 2019 and April 2021. Results: Of 2119 cancer patients surveyed, the number of GC patients was 614 (29.0%). Out of 614 GC patients, 39 (6.4%) had FGFR2 amplification alone, 22 (3.6%) had MET amplification, and 2 GC patients (0.3%) had concurrent FGFR2 and MET amplification. Two patients with concurrent FGFR2 and MET amplification did not respond to first-line chemotherapy. These two patients had significantly shorter overall survival (3.6 months) compared to patients with FGFR2 or MET amplification alone (13.6 months and 8.4 months, respectively) (*p* = 0.004). Lastly, we tested the existence of FGFR2 and MET in tumor specimens from different organ sites. Initially, the NGS was tested in a primary tumor specimen from stomach cancer, where the MET copy number was 14.1 and the FGFR2 copy number was 5.3. We confirmed that both MET and FGFR2 were highly amplified in the primary tumor using FISH (MET–CEP7 ratio = 5 and FGFR2–CEP7 ratio = 3). However, although the MET copy number was normal in peritoneal seeding using FISH, FGFR2 remained amplified using FISH (FGFR2–CEP7 ratio = 7) with high FGFR2 protein overexpression. Hence, there was intra-patient molecular heterogeneity. Conclusions: our findings suggest that concurrent amplification of FGFR2 and MET in GC patients is associated with clinical aggressiveness and may contribute to non-responsiveness to chemotherapy or targeted therapy.

## 1. Introduction

Gastric cancer is one of the most common malignancies of the digestive tract. Gastric cancer results in 1.1 million new cases and causes 770,000 deaths worldwide, making it the fifth most common cancer and one of the leading causes of cancer-related deaths [1]. Although early-stage gastric cancer patients have the chance of a cure through surgical resection, the majority of patients are diagnosed with inoperable disease status or experience disease recurrence. For such patients, systemic chemotherapy was the main therapeutic option for palliative treatment. However, the treatment outcome for advanced stage or recurrent gastric cancer remains poor, with a median overall survival of 12 months and a 5-year survival rate of less than 10%, despite several lines of systemic chemotherapy [2,3]. The poor response to conventional chemotherapy and the unfavorable prognosis of gastric cancer underscore the urgency of developing new therapeutic options. Recent advancements in molecular biology have led to innovative approaches in the treatment of gastric cancer.

Receptor tyrosine kinases (RTKs) are a protein family that play a pivotal role in cell proliferation, migration, and death [4]. Genetic alterations in RTKs have been found in various tumors and are considered to contribute to tumorigenesis through downstream MAPK/PI3K/AKT signaling pathways [5,6,7,8]. In particular, a subset of RTK alterations including human epithelial growth factor receptor-2 (HER-2), mesenchymal epithelial transition factor receptor (MET), and fibroblast growth factor receptor 2 (FGFR2) are found in a subset of gastric cancer patients, and are associated with advanced tumor condition, metastasis, and poor prognosis of gastric cancer patients [9,10]. Considering the low efficacy of chemotherapy in gastric cancer, the therapeutic targeting of genetic alterations in RTKs has garnered interest in patients with these features. HER-2-targeted therapy in well-selected HER-2-positive (IHC—Immunohistochemstry—3 positive or 2 positive with FISH—fluorescence in situ hybridization—positive) gastric cancer has successfully improved the survival of such patients in a phase III trial [11]. However, targeting other types of RTKs such as MET and FGFR therapy has failed to show clinical benefit [12,13,14,15]. One possible explanation for this failure is the presence of various genetic alterations affecting downstream pathways of RTKs. A good example is the co-existence of genetic alterations in the same RTK family. While genetic alterations in related RTKs are generally known to be mutually exclusive [16,17], recent studies conducted using more advanced technical methods have reported contradictory findings [18,19]. As a result, further studies with a large cohort and precise methods for detecting genetic alterations, such as large panel NGS, would be warranted.

With the implementation of NGS in the clinic, genomic alterations in the tumor specimen are available to clinicians to guide patients into clinical trials. Understanding the landscape and prevalence of each genomic aberration in real-world data is very important, especially when developing drugs for each target. Since NGS has been implemented in 2017 as part of clinical practice in Korea, a rare subset of genomic aberrations such as rare fusions and/or amplifications has been defined in our clinical data set [20,21]. Furthermore, the existence of concurrent mutations and/or amplifications may complicate the medical decision-making process such as the sequence of targeted agents or combination of the targeted agents to optimize the treatment outcome. Hence, it is important to analyze the clinical implication of rare concurrent amplifications to optimize treatment strategies.

In the preliminary investigation using RNAseq data of gastric cancer patients’ tissue, we found that the combined evaluation of RTKs, such as *MET* and *FGFR2* amplification, could identify the specificity of gastric tumor cells when compared to normal tissue. As a follow-up to this, we conducted additional analyses to assess the relationship between two genes using real-world NGS data from gastric cancer patients. We aimed in the current study to find the frequency of *MET* and *FGFR2* amplification and their impact on gastric cancer treatment based on real-world data. We also emphasized the importance of tumor heterogeneity, as each genomic aberration was present in different organs, making it difficult to optimize gastric cancer treatment.

## 2. Materials and Methods

### 2.1. Patient Enrollment

We collected the results of the NGS test, treatment outcomes, and clinicopathologic characteristics of cancer patients who underwent the NGS test as a part of routine practice in Samsung Medical Center from 2019 to 2021. Treatment outcome variables include the rest response of chemotherapy, the objective response rate, progression-free survival, and overall survival. Clinicopathologic characteristic variables include sex, age at diagnosis, type of chemotherapeutic agent, tumor mutational burden (TMB) status (≥10 mutations/mb vs. <10 mutations/mb), microsatellite instability (MSI) status (MSS vs. MSI high), and programmed cell death ligand1 (PD-L1) combined positive score (CPS 0 vs. ≥1).

The collection of specimens and associated clinical data used in this study was approved by the Institutional Review Board of Samsung Medical Center (IRB File No. 2021-09-052). All patients who participated in this study provided written informed consent prior to enrollment and specimen collection. This study was performed in accordance with the principles of the Helsinki Declaration and the Korean Good Clinical Practice guidelines.

### 2.2. DNA Extraction

For the majority of tumor tissues, tumor areas were micro-dissected, with the exception of samples designated for genomic DNA extraction. From formalin-fixed paraffin-embedded (FFPE) tissue pieces, genomic DNA was extracted and then cleaned up with the AllPrep DNA/RNA FFPE Kit from Qi-agen (Venlo, The Netherlands). Using the Qubit dsDNA HS assay kit from Thermo Fisher Scientific in Waltham, MA, USA, the DNA concentration was ascertained. For the library setup, 40 ng of DNA was utilized. The DNA integrity score, indicative of the DNA fragment length and thus its quality, was gauged through the Genomic DNA ScreenTape test using the Agilent 2200 TapeStation device from Agilent Technologies in Santa Clara, CA, USA.

### 2.3. Library Preparation and Data Analysis

The DNA library was assembled using the TruSight Oncology 500 DNA/RNA NextSeq Kit (Illumina, San Diego, CA, USA) based on hybrid capture, adhering to the kit’s guidelines. The enrichment chemistry was fine-tuned during this process to effectively capture nucleic acids from FFPE samples. In the TruSight Oncology 500 (TSO 500) evaluation, Unique Molecular Identifiers (UMIs) were employed to ascertain distinct coverage at every point and minimize the interference from sequencing errors and FFPE sample deamination. This method in DNA library construction enhances the detection of low-frequency variants while concurrently reducing inaccuracies, leading to heightened specificity.

Clinically significant genomic changes, such as SNVs, small insertions/deletions (indels), CNVs, and rearrangements/fusions were assessed from the sequence data. SNVs and minor indels showing a variant allele frequency (VAF) below 2% were disregarded. Average CNVs exceeding four were viewed as a gain, while those below one signified a loss. In the TSO 500-CNV analysis, only amplifications (gains) were examined. The TSO 500 software (Local App version 1.3.0.39.) from Illumina in San Diego, CA, USA was employed to determine TMB and MSI states. TMB evaluation involved: (1) ignoring any variant seen ≥10 times in databases like GnomAD exome, genome, and 1000 genomes; (2) including changes in the coding region based on RefSeq Cds; (3) variants with a frequency ≥ 5%; (4) those with coverage ≥ 50X; (5) encompassing SNVs and indels; (6) both nonsynonymous and synonymous changes; and (7) excluding certain nonsynonymous and synonymous changes. The applicable panel scope for TMB corresponds to the entire coding area with coverage over 50X. MSI was computed using microsatellite markers, gauging instability against a reference group of standard samples, founded on information entropy measures. The fraction of unstable MSI markers from the complete evaluated MSI markers was documented as an individual sample’s microsatellite score.

### 2.4. Sequencing of Whole Transcriptome

We determined the concentration and quality of total RNA using the Quant-IT RiboGreen method (from Invitrogen, Waltham, MA, USA). Post analysis on the TapeStation RNA ScreenTape from Agilent, we utilized 100 ng of the total RNA to create a sequencing library, adhering to the instructions provided with the TruSeq RNA Access Library Prep Kit from Illumina. Once the total RNA was fragmented, the fragments were transcribed into first-strand cDNA using SuperScript II reverse transcriptase (from Invitrogen) and an array of random primers, which was followed by second-strand cDNA creation. Post purification, PCR was employed to enrich the products, forming the cDNA library. After standardizing and merging six libraries into one hybridization/capture reaction, these pooled libraries were exposed to biotin-tagged oligos that matched the genome’s coding sections. Targeted library sequences were then captured using oligo probes marked with biotin and attached to streptavidin-linked beads. To quantify the assembled libraries, we used the KAPA Library Quantification kits tailored for Illumina Sequencing systems, in line with the qPCR Quantification Protocol Guide (offered by KAPA BIOSYSTEMS, reference #KK4854). These indexed libraries were later processed on an Illumina HiSeq2500 instrument. The paired-end sequencing task was overseen by Macrogen Inc., based in Seoul, South Korea. The RNA sequence annotations utilized ENSEMBL (version 98), and alignment to the human reference genome (GRCh38) was achieved with STAR software [22]. Transcripts were quantified in TPM (transcript per million) units using the RSEM method (version 1.3.1) [23]. Any TPM readings below one were treated as null values.

### 2.5. Immunohistochemistry (IHC) Test

We used the same method from our previous reports [20,21]. Briefly, the FGFR2 IHC test was performed using Benchmark XT (Ventana, Tucson, AZ, USA). Fixed tissues were embedded in paraffin blocks and sectioned at 3 μm thickness. Each section was deparaffinized in xylene, and antigen retrieval was performed. Samples were incubated with anti-FGFR2 using a Dako Autostainer Link 48 (Agilent Technologies, Santa Clara, CA, USA). All IHC samples were scanned using a ScanScope Aperio AT Turbo slide scanner (Leica Microsystems, Melbourne, Australia).

### 2.6. Statistical Analysis

We analyzed the association between *FGFR2* and/or *MET* amplification with treatment outcome. The tumor response evaluation was categorized into complete response (CR), partial response (PR), stable disease (SD), and progressive disease (PD). The objective response rate (ORR) was defined as the proportion of patients who achieved a CR or PR based upon the best response. Overall survival (OS) was calculated from the start of chemotherapy until death or the last follow-up. Progression-free survival (PFS) was calculated from the start of chemotherapy until disease progression, death without documented progression, or the last follow-up. Survival was calculated using the Kaplan–Meier method and survival curves were compared using the rog-rank test. Correlations between *FGFR2/MET* amplification and other clincopathologic variables including tumor mutational burden status, microsatellite instability status, and PD-L1 CPS score were estimated using Pearson’s correlation analysis. Data are presented as the mean ± SD. All statistical analyses were performed using R (Ver.3.4), R studio (https://www.rstudio.com/). Statistical significance was set at *p* < 0.05. All statistical tests were two-sided.

## 3. Results

### 3.1. Prelimnary Analysis Choosing FGFR2 and MET Amplification

In the preliminary investigation phase, we conducted an analysis using RNAseq to explore the expression trends of FGFR2 and MET. Initially, we determined the extent of their overexpression in tumors compared to normal tissues through assessing the log2 fold change values. Within the same samples, we then compared the log2 fold change values of these two genes (Appendix A). Interestingly, in instances where the log2 fold change value of FGFR2 was low, MET showed a high log2 fold change value, and vice versa. Even though individual evaluations of FGFR2 and MET might not indicate significant differences between normal and tumor samples, a combined assessment of the expression levels of both genes revealed statistically significant disparities (Appendix A).

### 3.2. Patient Characteristics

A total of 2119 cancer patients received next-generation sequencing using a panel targeting 523 cancer genes (TSO500, Illumina) as part of clinical practice at Samsung Medical Center between 2019 and 2021. Among 2119 patients, there were 614 (29.0%) patients with gastric cancer. Among gastric cancer patients, 39 (6.4%) patients had *FGFR2* amplification, and 22 (3.6%) patients had *MET* amplification. Although most patients had mutually exclusive *FGFR2* or *MET* amplifications, two (0.3%) gastric cancer patients had both FGFR2 and MET amplifications detected in the same tumor specimen using NGS (Figure 1A).

Next, we analyzed the distribution of copy numbers for each gene (Figure 1B). The copy numbers of *FGFR2* tended to be higher than those of *MET* (*p* = 0.03, median values: *FGFR2* (19.6) vs. *MET* (9.6)) (Figure 1B). In the two patients who had *FGFR2* and *MET* amplification simultaneously, their copy numbers of *FGFR2* were 3.7 and 5.1, and the copy numbers of *MET* were 14.1 and 5.3.

In terms of treatment response to first-line chemotherapy, the two patients with *FGFR2* and *MET* amplification showed significantly worse outcomes than those with either gene amplification alone (Figure 1C). The overall survival (OS) of patients with both *FGFR2* and *MET* amplification was significantly shorter than patients with *FGFR2* amplification only or *MET* amplification only (*p* = 0.004, median values: 380 days (FGFR2 amplification only), 260 days (MET amplification only) vs. 102 days (both FGFR2 and MET amplification)) (Figure 1D).

Next, we evaluated the correlation between *FGFR2/MET* amplification and tumor mutational burden (TMB) status (≥10 mutations/mb vs. <10 mutations/mb), microsatellite instability (MSI) status (MSS vs. MSI high), and PD-L1 CPS score (CPS 0 vs. ≥1) (Figure 1E). Among the 59 patients, only 3 patients had high TMB (≥10 mutations/Mb), and all patients had MSS (Figure 1E). The most frequently co-amplified genes were *MYC* (n = 12, 20.3%), followed by *FGF19* (n = 9, 15.3%), *FGF3* (n = 8, 13.6%), *CCND1* (n = 8, 13.6%), and *RICTOR* (n = 7, 11.9%) (lower left panel of Figure 1E). Interestingly, among patients with both *FGFR2* and *MET* amplification, one patient did not have any other amplified genes or mutations, while the other had three amplified genes (*RICTOR, ERBB2,* and *CDK6*) (lower right panel of Figure 1E). In summary, these results suggest that the presence of amplification in both *FGFR2* and *MET* could be an important and influential factor in the response to chemotherapy and overall survival.

At last, we made an effort to validate the findings of our study in other gastric cancer cohorts with an NGS data set. We tried to find the concurrent amplification of FGFR2 and MET in gastric cancer through a search in public cohort data (cbioportal.org), but we were unable to find it (Appendix A).

Of note, a 43-year-old man with gastric cancer and peritoneal seeding, who had MET amplification, showed a mixed response to the MET inhibitor. Following the treatment, the primary gastric lesion improved, but ascites persisted without improvement. We tested the existence of FGFR2 and MET in tumor specimens from different organ sites. Initially, the NGS was tested in a primary tumor specimen from stomach cancer, where the MET copy number was 14.1 and the FGFR2 copy number was 5.3. We performed MET FISH, MET IHC, FGFR2 FISH, and FGFR2 IHC on both primary tumor and peritoneal seeding specimens (Figure 2). Of note, we confirmed that both MET and FGFR2 were highly amplified in the primary tumor using FISH (MET–CEP7 ratio = 5 and FGFR2–CEP7 ratio = 3) (Figure 2 upper panel). However, although the MET copy number was normal in peritoneal seeding using FISH, FGFR2 remained amplified using FISH (FGFR2–CEP7 ratio = 7) with high FGFR2 protein overexpression.

## 4. Discussion

In the current study, we analyzed the results of the NGS test of 614 gastric cancer patients and their clinical outcomes. *MET* amplification was identified in 22 (3.6%) patients, and *FGFR2* amplification was identified in 39 (6.4%) patients based on tissue specimen NGS analysis. These incidence findings were consistent with previous studies, with MET amplification reported in 4–10% of cases and *FGFR2* amplification in 4–15% of gastric cancer cases [9,10]. Regarding survival according to genetic amplification, the overall survival (OS) for patients with *FGFR2* amplification was 13.6 months, and for patients with MET amplification was 8.4 months. Numerous studies consistently link *MET* and *FGFR2* amplification with advanced tumor stages, metastasis, and poorer survival outcomes in gastric cancer [9,10,24]. Our study findings are in line with previous research, reaffirming the prognostic significance of *MET* and *FGFR2* amplification in gastric cancer.

We have encountered a rare subset of gastric cancer patients with both *FGFR2* and *MET* amplification on the same tumor specimen. In the current study, two (0.3%) patients showed co-amplification of both *FGFR2* and *MET*. To the best of our knowledge, a subset of patients with concurrent amplifications of oncogenic genes had not been reported, especially in FGFR2/MET for gastric cancer patients. Although NGS is rapidly being implemented in clinical practice globally, various alterations per patient and per tumor type are being reported in the clinic. In addition, various NGS panels with a larger number of genes are being developed and used in the clinic. Hence, the frequency of encountering rare genomic aberrations is increasing in the oncology clinic.

Of note, patients with co-amplification of *EGFR2* and *MET* showed disastrously poor survival outcomes of 3.6 months despite several lines of chemotherapy. Co-amplification of RTKs has a synergistic effect on tumor progression through shared downstream signaling pathways [25]. In in vitro studies, the co-stimulation of EGF and HGF (hepatocyte growth factor) showed a synergistic effect on cell proliferation in cancer cells expressing both MET and EGFR (epidermal growth factor receptor) [26]. Furthermore, MET and EGFR receptors can be trans-activated through interactions mediated through transphosphorylation upon activation of other receptors, without the need for ligand-mediated receptor activation [6,27]. In our study, we found that individual evaluation of *MET* and *FGFR2* did not show significant differences between tumor and normal tissue samples, but combined assessment of both genes revealed significant amplification in tumor samples than normal tissue based on RNAseq preliminary data (Appendix A). Our study findings support the previous studies that suggest co-amplification of RTKs promotes tumor cell progression. The interactions between RTKs may contribute to resistance to targeted therapy. Preclinical data support the notion that co-inhibition of MET and FGFR2 enhances or restores the anti-tumor effect of FGFR2 inhibitors in co-amplified cancer cells [28,29,30]. Similar findings have been observed in clinical trials involving patients with concurrent *MET* and *HER-2* amplification [31,32]. Collectively, these findings suggest that concurrent amplification of *FGFR2* and *MET* is a major factor influencing the response to chemotherapy and overall survival.

Following the success of HER-2-targeted therapy in HER-2 positive gastric cancer, several monoclonal antibodies and small molecules targeting other RTKs such as MET and FGFR2, along with their downstream signaling pathways, have been investigated [12,13,14,15,33,34]. However, the majority of these studies have failed to demonstrate clinical benefit. As mentioned, technological advancements enable the detection of genetic mutations to an extent previously unattainable and provide a higher level of reliability. Accurate detection of gene amplification is a crucial step in selecting eligible patients and predicting treatment response to targeted therapy. Recently, we demonstrated that *MET*-amplified gastric cancer patients without co-amplification of other RTKs can benefit from *MET* inhibitors compared to standard second-line chemotherapy, based on NGS results from tissue specimens [35]. Jogo et al. reported that some gastric cancer patients with *FGFR2* amplification detected only through NGS based on ctDNA showed tumor responses to FGFR inhibitors [18]. Conversely, Janjigian et al. reported that a subset of HER-2-positive esophagogastric cancer patients identified as positive through IHC/FISH showed discordant results when analyzed using NGS, leading to significantly worse progression-free survival on HER-2-targeted therapy [36]. These findings emphasize the significance of precise genetic alteration detection methods in patient selection.

Another critical factor contributing to the success of targeted therapy is inter-tumoral molecular heterogeneity. In our study, when *FGFR2* and *MET* were confirmed at different sites (i.e., primary and peritoneum), we demonstrated each organ had different genomic alterations. This patient was enrolled on to a MET inhibitor (small molecule) trial, and as implicated in genomic alterations, his stomach cancer was responsive to the MET inhibitor, but his peritoneal seeding was rapidly deteriorating, which resulted in death from the disease. This observation of intra-patient molecular heterogeneity aligns with previous reports [32,35,36,37,38]. The main challenge in the clinical practice of identifying multiple genetic mutations is the limitation of available tissue samples.

To overcome the challenges posed by tumor heterogeneity, we believe that circulating tumor DNA (ctDNA) analysis could provide a solution. The clinical utility of cell-free NGS may be an alternative or complementary method to genomically characterize a patient’s tumor since it may reflect the summation of shedding tumor DNA [39]. In line with our observation, a recent Japanese study identified FGFR2 amplifications using ctDNA in gastric cancer patients that were not detected using tissue NGS, and these patients showed a response to FGFR inhibitors. In their report, ctDNA analysis also revealed the presence of *MET* co-amplification, which explained the lack of response to FGFR inhibitors in a patient with *FGFR2* amplification [18]. Another large study involving ctDNA-based NGS in gastroesophageal cancer reported that the identification of *HER-2* or *EGFR* amplification using ctDNA was associated with increased patient enrollment in clinical trials and improved survival outcomes through targeted inhibition [40].

Detecting as many genetic alterations as possible not only expands the pool of candidates for targeted therapy but also enables the anticipation of potential resistance mechanisms, and the induction of anticancer responses and improved survival rates through strategies such as multiple targeted therapies. Considering the challenges associated with repeated biopsies of metastatic lesions, ctDNA and NGS should be considered as a screening method for tailoring targeted therapy in gastric cancer, providing valuable information on tumor heterogeneity and guiding treatment decisions.

## 5. Conclusions

Our observation underscores the importance of understanding concurrent genomic aberrations per patient and per tumor types. For medical decisions, a more tailored decision-making process needs to be considered such as excluding patients with several concurrent aberrations for clinical trials or re-biopsy at the time of clinical trial enrollment. In addition, ctDNA NGS can also be useful to guide patients for optimized treatment in future trials. Further studies are warranted to investigate the clinical implications of genetic alteration heterogeneity and explore potential therapeutic strategies that could improve outcomes for such patients.

## Figures and Tables

**Figure 1 biomedicines-11-03172-f001:**
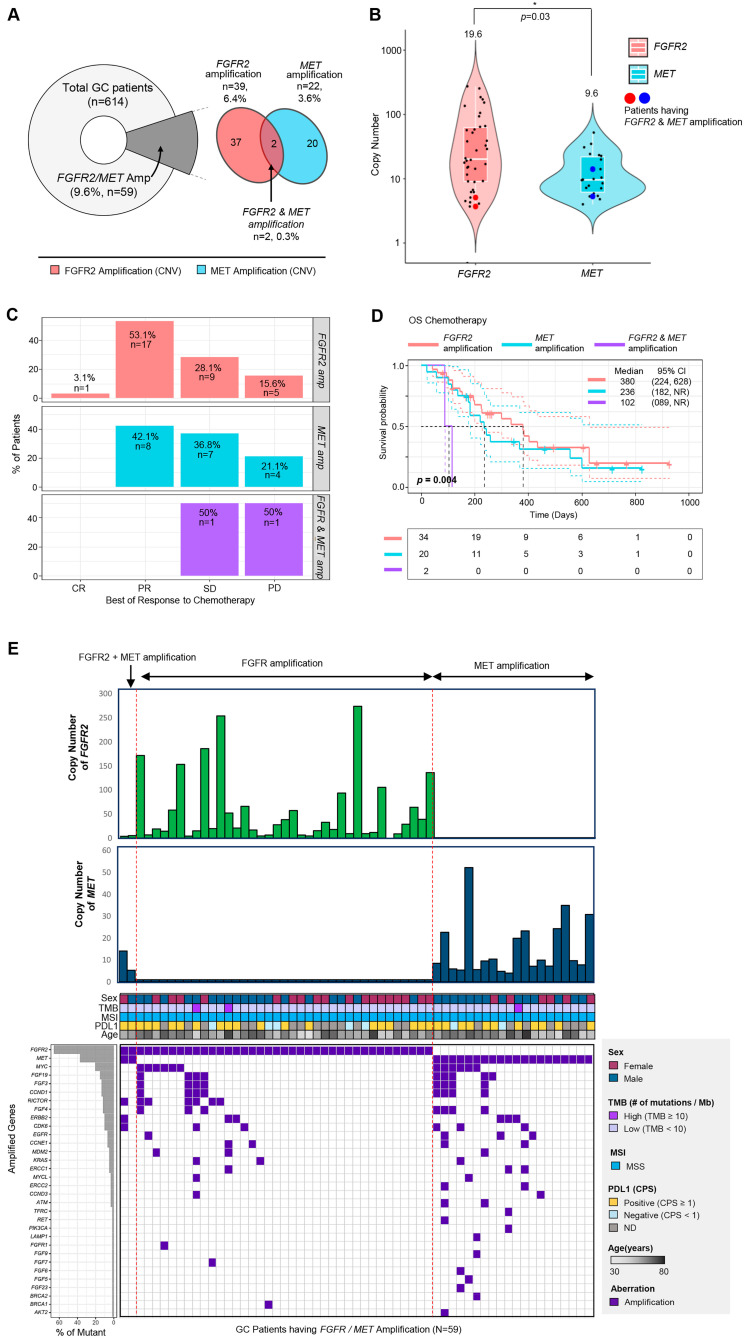
Overview of enrolled cancer patients and the proportions of MET and FGFR2 genetic variants in gastric cancer patients. (**A**) A pie chart representing the proportion of patients having FGFR2 or MET amplification (left), and a Venn diagram showing the number and percentage of the patients with *FGFR2* amplification and *MET* amplification. (**B**) The range of copy number in each gene (*FGFR2* and *MET*). The middle white line of the boxplot represented the mean value of copy numbers. Copy numbers of two patients having both *FGFR2* and *MET* amplification were dotted as the large red or blue dot. (**C**) The percentage of patients in each best of response category (CR, PR, SD, PD) to chemotherapy depending on their amplified gene type (*FGFR2* only, *MET* only, *FGFR2* and *MET*). (**D**) Overall survival curve of the patients with *FGFR2* and/or *MET* amplification (red line, *FGFR2*-only amplification; blue line, *MET*-only amplification; purple line, *FGFR2* and *MET* amplification). (**E**) Comprehensive landscape of gastric cancer patients with *FGFR2* and/or *MET* amplification with other clinical factors (copy number, sex, TMB, MSI, PD-L1, age). Co-amplified genes were represented as a purple box (lower part of Figure 1E).

**Figure 2 biomedicines-11-03172-f002:**
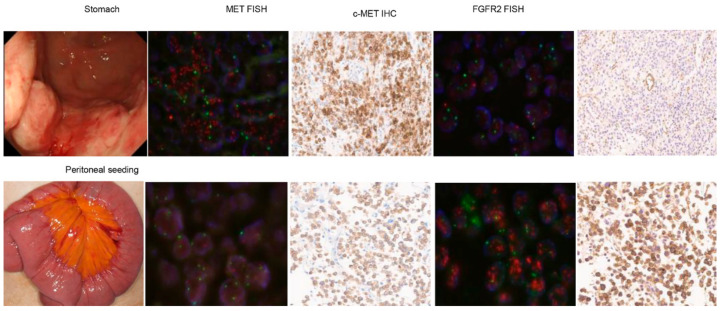
Evaluation of treatment outcome and MET/FGFR gene status in patients with concurrent MET and FGFR amplification. Initially, the NGS was tested in a primary tumor specimen from stomach cancer, where the MET copy number was 14.1 and the FGFR2 copy number was 5.3. We performed MET FISH, MET IHC, FGFR2 FISH, and FGFR2 IHC on both primary tumor and peritoneal seeding specimens (Figure 2). Of note, we confirmed that both MET and FGFR2 were highly amplified in the primary tumor using FISH (MET–CEP7 ratio = 5 and FGFR2–CEP7 ratio = 3) (Figure 2 upper panel). However, although the MET copy number was normal in peritoneal seeding using FISH, FGFR2 remained amplified using FISH (FGFR2–CEP7 ratio = 7) with high FGFR2 protein overexpression.

## Data Availability

Data are contained within the article, Appendix A, and European Nucleotide Archive (Accession number: PRJEB69296).

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
