# Peer review of "Clinical Implication of Concurrent Amplification of MET and FGFR2 in Metastatic Gastric Cancer"

_biomedicines, 2023, doi:10.3390/biomedicines11123172_

Round 1
Reviewer 1 Report
Comments and Suggestions for Authors
In the present manuscript, it suggested that concurrent amplification of FGFR2 and MET in gastric cancer patients is associated with clinical aggressiveness such as non-responsiveness to chemotherapy. However, the manuscript is not well-written. I recommend that this paper be accepted after minor revision.
1. In Figure 1E, it is difficult to find the patients with FGFR and/or MET amplification because the bars are low bar.
2. In line 194-197, it mentioned that one patient did not have any other amplified genes, and one patient had only 1 amplified gene (CDK6). However, the one patient has 3 amplified genes in Figure 1E. These are no CDK6 gene in amplified gene list.
3. Almost half of patients with FGFR or MET amplification do not have any other amplified genes in Figure 1E. Do the patients with FGFR or MET amplification have other amplified genes except Figure 1E?
4. In line 128, the reference may be different because the article did not use the TSO 500 pipeline.

Reviewer 2 Report
Comments and Suggestions for Authors
well written study.
1- please define aim
2-(https://doi.org/10.1007/s00268-016-3858-6) and (https://doi.org/10.1016/j.cireng.2015.02.004) suggested studies for the references
Comments on the Quality of English Languagegood
Reviewer 3 Report
Comments and Suggestions for Authors
The authors analyzed the clinical outcomes of GC patients with MET and FGFR2 amplification using the next-generation sequencing (NGS) database cohort at Samsung Medical Center. They found that concurrent amplification of FGFR2 and MET in GC patients is associated with clinical aggressiveness and may contribute to non-responsiveness to chemotherapy or targeted therapy. There are several problems:
1. Is the data publicly available? Can it be uploaded onto public database? Otherwise, how to make sure such analysis or results are reproducible?
2. Why choose MET and FGFR2 for concurrent study? The authors need to justify the choice of these two genes. Are there other genes studied or worth to be studied?
3. The authors need to validate their findings on an independent dataset.
4. The authors need to do more functional analysis of how MET and FGFR2 works and add a mechanism figure.
Comments on the Quality of English LanguageThe authors analyzed the clinical outcomes of GC patients with MET and FGFR2 amplification using the next-generation sequencing (NGS) database cohort at Samsung Medical Center. They found that concurrent amplification of FGFR2 and MET in GC patients is associated with clinical aggressiveness and may contribute to non-responsiveness to chemotherapy or targeted therapy. There are several problems:
1. Is the data publicly available? Can it be uploaded onto public database? Otherwise, how to make sure such analysis or results are reproducible?
2. Why choose MET and FGFR2 for concurrent study? The authors need to justify the choice of these two genes. Are there other genes studied or worth to be studied?
3. The authors need to validate their findings on an independent dataset.
4. The authors need to do more functional analysis of how MET and FGFR2 works and add a mechanism figure.
Round 2
Reviewer 3 Report
Comments and Suggestions for Authors
The authors need to upload the NGS data onto a publicly available database. The independent validation results should be added to the main manuscript.
